# Beyond Slow Signs in High-fidelity Model Extraction

**Hanna Foerster and Robert Mullins**
University of Cambridge

**Ilia Shumailov and Jamie Hayes**
Google DeepMind

## Abstract

Deep neural networks, costly to train and rich in intellectual property value, are increasingly threatened by model extraction attacks that compromise their confidentiality. Previous attacks have succeeded in reverse-engineering model parameters up to a precision of `float64` for models trained on random data with at most three hidden layers using cryptanalytical techniques. However, the process was identified to be very time consuming and not feasible for larger and deeper models trained on standard benchmarks. Our study evaluates the feasibility of parameter extraction methods of Carlini et al. [1] further enhanced by Canales-Martínez et al. [2] for models trained on standard benchmarks. We introduce a unified codebase that integrates previous methods and reveal that computational tools can significantly influence performance. We develop further optimisations to the end-to-end attack and improve the efficiency of extracting weight signs by up to 14.8 times compared to former methods through the identification of easier and harder to extract neurons. Contrary to prior assumptions, we identify extraction of weights, not extraction of weight signs, as the critical bottleneck. With our improvements, a 16,721 parameter model with 2 hidden layers trained on `MNIST` is extracted within only 98 minutes compared to at least 150 minutes previously. Finally, addressing methodological deficiencies observed in previous studies, we propose new ways of robust benchmarking for future model extraction attacks.

## 1 Introduction

Training machine learning (ML) models requires not only vast datasets and extensive computational resources but also expert knowledge, making the process costly. This makes models lucrative robbery targets. The rise of ML-as-a-service further amplifies the challenges associated with balancing public query accessibility and safeguarding model confidentiality. This shows the emerging risk of model extraction attacks, where adversaries aim to replicate a model's predictive capabilities from a black box setting where only the input and output can be observed. Previous attacks have approached model extraction either precisely, obtaining a copy of the victim model, or approximately, obtaining an imprecise copy of the victim model. Components of the model that have been the target of extraction include training hyperparameters [3], architectures [4] [5] [6], and learned parameters such as weights and biases in deep neural networks (DNNs) [1] [7].

In this paper, we are interested in precise model extraction and use the most recent advances of cryptanalytical extraction of DNNs by Carlini et al. [1] and Canales-Martínez et al. [2] as our starting point. Carlini et al. previously demonstrated that it is feasible to extract model *signatures* – normalised weights of neural networks – on relatively small models with up to three hidden layers. For 1 hidden layer models the parameter count tested was up to 100,480, however, for 2 and 3 hidden layer models, which are increasingly difficult to extract, the maximum parameter count for models tested was 4,020. While signature extraction was generally determined as straightforward, *sign* extraction of the weights was identified as a bottleneck with further work needed. Following this, Canales-Martínez et al. [2] improved sign extraction speed from exponential to polynomial time, with their Neuron

Wiggle method. However, Canales-Martínez et al. stopped short of evaluating the full extraction pipeline, focusing only on the signs.

In our work, we perform a deeper performance evaluation of the signature and sign extraction methods. We first create a comprehensive codebase that integrates Carlini et al. [1]'s signature extraction technique with the sign extraction technique of Canales-Martínez et al. [2], allowing for systematic and fair benchmarking. We identify inefficiencies in the combined *signature–sign* method interactions and discover that we can significantly improve on the end-to-end attack efficacy. Importantly, we find that Canales-Martínez et al.'s sign extraction already eliminates the sign extraction bottleneck observed in prior work. Further improving on sign extraction we speed up the process by up to $14.8$ times compared to Canales-Martínez et al., so that sign extraction only takes up as little as $0.002\%$ of the whole extraction time for larger models. For larger models due to randomness of the start of the search and the position of neurons in space, the signature extraction can take a long time, highlighting the importance of considering the full pipeline extraction compared to only the sign extraction. For smaller or less complex models, signature extraction is much faster, and then the sign extraction time becomes more significant. For these models we see on average that the whole parameter extraction process is sped up by about $1.2$ times and a speed up of up to $6.6$ times can be attained when quantizing some extraction sub-routines to `float32`.

We make the following contributions:

1. **Optimizing Extraction Strategies:** We modify the extraction process to only sign extract neurons requiring trivial effort, finding that spending more time in extracting harder to sign-extract neurons does not lead to higher success in correct sign extraction. This significantly reduces the number of queries needed. We find that these harder to sign-extract neurons' sign extraction can be pipelined with other operations, improving both robustness and speed of sign extraction. An additional deduplication process and a suggestion to quantize some sub-routines speeds up the overall extraction time.

2. **Redefining Bottlenecks in Extraction Processes:** By optimizing sign extraction, we find that, contrary to earlier studies, extraction is now dominated by signature extraction. This shifts the focus of optimization efforts for achieving scalable high-fidelity extraction.

3. **Addressing Methodological Shortcomings:** We determine that improvements reported in some prior work come from unfair comparison between (1) standard benchmarks and models trained on random data; (2) models trained with different randomness; (3) models extracted using different randomness; (4) models with more hidden layers or different sizes.

Our codebase can be found at https://github.com/hannafoe/cryptanalytical-extraction.

## 2 Related Work

The analysis of neural network model extraction strategies in Jagielski et al. [8] categorises three types of extraction attacks. *Functionally equivalent extraction* aims to replicate the target model's output for each input and is the most exact method of extraction. *Fidelity extraction* seeks to closely mirror the target model's output on specific data distributions, such as label agreement. *Task accuracy extraction* aims to match or surpass the target model's task performance without replicating errors. Fidelity and accuracy differ in that the goal for fidelity is close replication of the target model, where the target model is considered the benchmark label, whereas for task accuracy the goal is to closely mirror the ground-truth labels of the dataset.

### 2.1 Learning-based Methods: Fidelity and Task accuracy extraction

Learning-based methods that train a substitute model closely mirroring the target model are used for fidelity and task accuracy extraction. Since knowledge from the target blackbox model is transferred to a simpler model, architecture knowledge is not always necessary [9], but according to Oh et al. [3] is found to raise fidelity. Moreover, while the training dataset is often not assumed, many works such as Tramèr et al. [10] assume knowledge of the problem domain and training data statistics to recover data distributions, and it has been shown that models trained on non-domain data significantly underperform [9, 11]. For instance, Papernot et al. [12] train models on self-created datasets mimicking `MNIST`, while others use GANs or other methods to produce data for training (Oliynyk et al. [13], Truong et al. [14], Correia-Silva et al. [15]).

Jagielski et al. [8] discuss how learning-based methodologies face challenges due to various non-deterministic factors, such as the random initialization of model parameters, batch formation sequence, and the unpredictable behaviours in GPU processing. In a study where the adversary is assumed to have complete access to both the training data and hyperparameters, and where the original model is used as a labeling oracle to help train a replication model with the same parameters from scratch, $93.4\%$ was the maximum fidelity reached by the replicated model. This is attributed to the non-deterministic elements in the learning processes of both the original and replicated model. More recent work by Martinelli et al. [16] that suggests that high fidelity can actually be reached with learning based methods. However, they assume a slightly different setting, and this only works for smaller networks, and appears to be more expensive.

## 2.2 Cryptanalytical Methods: Functionally equivalent extraction

The fidelity limitations of learning-based methods have prompted the development of techniques based on side-channel analysis or cryptanalysis to precisely extract parameters and achieve functional equivalence. Notable methods, such as those introduced by Lowd and Meek [17], Milli et al. [18], Batina et al. [19], Jagielski et al. [8], while contributing to this area, exhibit limitations in handling standard benchmarks or are confined to neural networks with only two layers, limiting its practicality. Rolnick and Körding [7] and Carlini et al. [1], attempt to extend cryptanalytical attacks to deeper neural networks but face efficiency limitations in larger models. Side-channel techniques appear inefficient for parameter extraction and are primarily used for architecture extraction [4] [6], and so we focus on cryptanalytical parameter extraction continuing from Carlini et al. [1]'s work.

Carlini et al. [1] successfully extract model parameters from fully connected neural networks with precision up to `float64`, applicable even to deeper models with multiple hidden layers. Their tests show that a model with one hidden layer allows extraction of up to 100,480 parameters using $2^{21.5}$ queries, while a three-layer model with 1,110 parameters is extracted with $2^{17.8}$ queries. This technique requires that the DNN uses ReLU activations, that the weights are in high precision, and that the output logits can be obtained. Unlike learning-based methods, it does not rely on known training data since it uses queries generated from a normal distribution or calculated coordinates that expose network parameters. However, the extraction of deeper models has not been tested, and all tests are conducted on 'random' models trained on data randomly drawn from a normal distribution. Their extraction is divided into two parts which includes the extraction of weight values up to a multiple and the extraction of the sign of the weights. The authors also note that sign extraction, while requiring only a polynomial number of queries, requires an exponential amount of time. The approach requires an exhaustive search over $2^n$, where $n$ is the number of neurons in a layer [2].

Canales-Martínez et al. [2] improved the sign extraction to be polynomial time. They apply extraction to a model with 8 hidden layers with 256 neurons each trained on `CIFAR10`, claiming that the entire sign extraction only required 30 minutes on a 256-core computer. However, Canales-Martínez et al. did not test the signature extraction and assumed signature extraction time to be relatively insignificant. So, these claims only hold for the sign extraction part of parameter extraction.

In our work, we use models trained on `MNIST` and `CIFAR10`, as well as randomly generated data, to benchmark full parameter extraction. Prior works have focused only on partial parameter extraction [2] or full extraction on models trained on random data [1]. Considering metrics such as query and time efficiency, we analyze the performance impact of Canales-Martínez et al.'s sign extraction method and propose additional optimization strategies.

## 3 Methodology

In what follows we explain the background methodology of sign extraction in Section 3.1 and its inefficiencies in prior work in Section 3.2, describe improvements to it and analyse its effect on the whole parameter extraction in Sections 3.3, 3.4, and finally outline some other improvements in Section 3.5.

### 3.1 Background: Understanding Sign Extraction

Neurons are the most basic components of neural networks. They create decision boundary hyperplanes in the parameter space and our goal is to identify these. These boundaries are defined by

weights and the bias associated with the neuron and it is this hyperplane equation that ultimately decides which inputs activate a neuron. Each neuron contributes to the final output of the model by either activating or deactivating with some given input. Locating the decision boundaries involves two main steps: identifying the neuron's value and determining its sign. While the exact scale of the hyperplane's normal vector is not crucial (as the hyperplane can be identified up to a multiplicative factor), the sign is critical as it determines how it partitions the parameter space and on which side of the hyperplane the inputs fall. In other words, the orientation of the decision boundary hyperplane is determined by the sign of the neuron. In a DNN, the sign determines the outcome of the matrix multiplication involving the input and weight matrix and the subsequent application of the ReLU activation. Depending on the sign of a neuron, a positive outcome could change to negative and be turned into $0$ through the ReLU activation, turning off the neuron's contribution to the final model output. Carlini et al. [1] introduce the concept of a neuron's signature, a set of normalised weight ratios $\left(\frac{a_1}{a_1}, \frac{a_2}{a_1}, \ldots, \frac{a_{d_{i-1}}}{a_1}\right)$, where each $a_i$ is an individual weight. Each signature signifies the neuron's decision boundary up to sign. They, however, identify the subsequent extraction of the sign as the bottleneck of the whole parameter extraction process. This is why Canales-Martínez et al. [2] develop the *Neuron Wiggle* method which speeds sign extraction up to polynomial time. In the following, we describe this sign extraction method, in order to describe our contributions on top of this. Please refer to Carlini et al. [1] and Canales-Martínez et al. [2] for an explanation of the whole parameter extraction routine that includes signature extraction.

**Neuron Wiggle Sign Recovery:** A wiggle at layer $i$ is defined as a vector $\delta \in \mathbb{R}^{d_{i-1}}$ of perturbations, where $d_{i-1}$ is the dimension of layer $i - 1$. The preimage of this wiggle at the input layer is added to witnesses, i.e., inputs $\mathbf{x}$ that place a neuron $\eta$ on the decision boundary of being activated or deactivated. At these 'critical points' on the decision boundary, the output of the neuron is $0$ and hence the effect of the perturbation on the neuron can be isolated. We define $\hat{A}^i$ to be the extracted weight matrix in layer $i$ and $\hat{A}_k^i$ to be the $k$th row corresponding to the weights of neuron $k$. Then, when the perturbation is added to the witness, neuron $k$ of layer $i$ changes its value by $e_k = \langle \hat{\mathbf{A}}_k^i, \delta \rangle$, i.e., the dot product of the extracted weights of neuron $k$ with the wiggle. Say the vector $\mathbf{c} = \{c_1, ..., c_{d_i}\}$ expresses the output coefficients of $\mathbf{x}$ in layer $i$. Then the difference in output that is induced by the wiggle can be expressed as $\sum_{k \in \mathbf{I}} c_k e_k$, where $\mathbf{I}$ contains all indices of active neurons in layer $i$.

We compute the maximal wiggle that affects only one neuron by aligning the wiggle $\delta$ parallel to $\hat{\mathbf{A}}_j^i$. This alignment ensures that the absolute value of the dot product $|\langle \hat{\mathbf{A}}_j^i, \delta \rangle|$ is maximised, making $\delta$ effective in activating the target neuron $j$ while minimizing impact on others. The reason for this is because $\delta$ being parallel to $\hat{\mathbf{A}}_j^i$ has either the same sign or opposite sign to $\hat{\mathbf{A}}_j^i$. So then, all summands in the dot product will have the same sign, preventing cancelling out each others effects and maximizing the summation. If no other row is a multiple of $\eta_j$ then the dot product of the row concerning $\eta_j$ with the wiggle should be bigger than the dot product of the wiggle with any other row of the extracted matrix. Overall, neuron wiggle sign recovery involves the following steps:

1. Project $\hat{\mathbf{A}}_j^i$ onto the orthogonal basis for the vector space at layer $i - 1$ and scale this projection to a small norm, $\varepsilon^{i-1}$, so that its influence in the output is minimal and only on the targeted neuron.

2. Calculate the preimage of this vector $\delta$ to obtain a wiggle in the input dimension $(F^{(i-1)})^{-1}(\delta) = \triangle$.

3. Calculate $f(\mathbf{x}^*)$, $f(\mathbf{x}^* + \triangle)$, $f(\mathbf{x}^* - \triangle)$.

4. Examine whether addition or subtraction of the wiggle activates the neuron.
   a. If adding $\triangle$ increases the magnitude of the output more than subtracting it, i.e., $|f(\mathbf{x}^* - \triangle) - f(\mathbf{x}^*)| < |f(\mathbf{x}^* + \triangle) - f(\mathbf{x}^*)|$, then $\eta_j$ is positive. So, $\hat{\mathbf{A}}_j^i = \mathbf{A}_j^i$.
   b. Conversely, if subtracting $\triangle$ results in a higher magnitude of the output difference, i.e., $|f(\mathbf{x}^* - \triangle) - f(\mathbf{x}^*)| > |f(\mathbf{x}^* + \triangle) - f(\mathbf{x}^*)|$, then $\eta_j$ is negative. So, $-\hat{\mathbf{A}}_j^i = \mathbf{A}_j^i$

In this paper, we focus on further reducing $s$, the critical parameter affecting sign extraction confidence and performance in Canales-Martínez et al. [2]'s method. Ultimately our optimisations reduce sign extraction to become an insignificant fraction of time and query effort of parameter extraction.

**Why is Confidence needed?** From the above explanation, the output effect of a target neuron is represented as the vector $c_j e_j$, where $c_j$ is the coefficient and $e_j$ is the effect of the target neuron. The com-

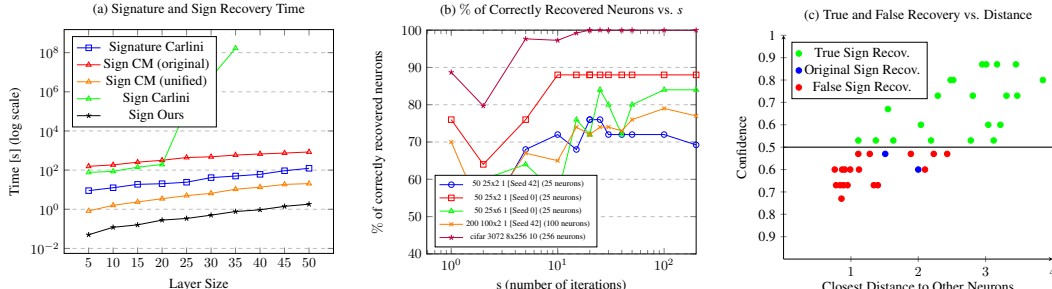

Figure 1: (a) Compares the running times for Carlini's signature extraction versus Carlini's sign extraction, Canales-Martinez (**CM**)'s sign extraction with $s = 200$ setting in the original implementation and in the unified implementation and Our sign extraction with $s = 15$ setting. The tests are across ten models with increasing layer sizes from $10-5-5-1$ to $100-50-50-1$, detailing times for a single layer's extraction in a non-parallelised setting. (b) Depicts how the average percentage of correctly recovered neurons in a layer changes when the number of sign extractions $s$ changes. Raising the number of sign extractions $s$ to more than 15 does not significantly raise the number of correctly recovered neurons. (c) Graph showing confidences of sign recovery when a hard neuron's euclidean distance to its neighbours is manipulated. These results are on hard to sign extract neurons 25 and 26 of an MNIST trained 784-32x8-1 model extracted with seed 42. The confidence metric scales from 1 to 0.5 first on the confidence of false sign recovery, which is equivalent to 0 to 0.5 of confidence in true sign recovery and then from 0.5 to 1 on the confidence of true sign recovery, resulting in the scale going from $1-0.5-1$.

bined output effect of all other active neurons $\in \mathbf{I}$ is expressed as $\sum_{k \in \mathbf{I} \setminus \{j\}} c_k e_k$. Incorrect sign recovery occurs under certain conditions: (**Necessary Condition**) the signs of $c_j e_j$ and $\sum_{k \in \mathbf{I} \setminus \{j\}} c_k e_k$ are opposite; (**Sufficient Condition:**) the magnitude of $c_j e_j$ is not significant compared to the magnitude of $\sum_{k \in \mathbf{I} \setminus \{j\}} c_k e_k$. This leads to a scenario where $\left| c_j e_j + \sum_{k \in \mathbf{I} \setminus \{j\}} c_k e_k \right| < \left| \sum_{k \in \mathbf{I} \setminus \{j\}} c_k e_k \right|$, meaning that the total activation output when the target neuron is active is less than the output when the target neuron is inactive. This arises when the effect of the target neuron's activation is overwhelmed by the opposing effects of the other neurons. Thus, we refer to "hard" neurons as those for which it is particularly challenging to compute a wiggle that isolates the neuron effectively.

To reliably determine the correct sign of the target neuron's effect, it is necessary that $|c_j e_j| > 2 \left| \sum_{k \in \mathbf{I} \setminus \{j\}} c_k e_k \right|$, which implies that the influence of the target neuron's wiggle must be greater than twice the aggregate effect of the other active neurons' wiggles. However, this criterion cannot be checked in practice because we would need to single out the contribution of each of the other neurons to the output given the witness plus perturbation as input. Yet, since the witness is not a witness to any of the other neurons, we cannot single out their effects. This is why Canales-Martínez et al. [2] introduce the concept of *confidence*. A series of $s$ different sign extractions on $s$ critical points and varying $\delta$ are conducted to gather diverse output responses. The *confidence* of this sign recovery is $s_{majority}/s$, where $s_{majority}$ is the number of sign extractions supporting the majority sign. The idea is that with more sign extractions, enough Neuron Wiggles will be constructed that are not overwhelmed by the opposing effects of other neurons, so that a confident decision can be made. In fact, according to Canales-Martínez et al., the problem of wrong signs extracted with the Neuron Wiggle method should be "fixable by testing more critical points" [2]. So, after testing on CIFAR10, they decide on fixing $s$ to 200 and recommend rerunning the Neuron Wiggle method on the least-confident 10% of sign recoveries.

### 3.2 Our Discoveries: Confidence In Practice

In practice, many neurons do not exhibit increased confidence even with additional iterations $s$. After an initially higher confidence on average due to not having seen enough samples, the confidence stabilises after about 10 iterations. Moreover, as illustrated in Figure 1(b), the number of correctly identified neurons does not increase beyond $s = 25$ for many models. Furthermore, sign recovery confidence varies notably. Neurons that are correctly recovered typically show a confidence level

above 0.75, while those incorrectly recovered average 0.64. This suggests that neurons with lower confidence are more likely to be incorrectly recovered, providing a practical indicator of recovery accuracy.

**'Easy' and 'Hard' to Sign Extract Neurons:** The idea that some neuron signs are "easier" to extract and some are "harder" to extract can be understood by picturing two neurons that are positioned very close to each other within a layer. Creating a "wiggle" that activates one without impacting adjacent neurons is difficult and becomes more complex as the number of tightly clustered neurons increases, hindering the ability to isolate and activate a single target neuron effectively.

In Figure 1(c), the change in confidence and true versus false sign recovery is depicted when a hard neuron's euclidean distance to its neighbours is manipulated. The blue dots depict the confidence of the two originally hard to extract neurons. To manipulate the target neuron to be further away from other neurons, Gaussian noise was added and to manipulate the target neuron to be closer to a cluster of the closest neurons, part of the distance to these closest neurons was added to make the target neuron the midpoint between closest neurons. One can see that if the target neuron is closer to other neurons, with high confidence the wrong sign will be recovered, whereas if the distance to other neurons is further, with high confidence the correct sign will be recovered.

**The Effect of an Incorrect Sign in the Extraction Process:** Prior extraction performance evaluations have primarily utilised models trained on random data originally developed by Carlini et al.. However, observations indicate that the percentage of correctly extracted neurons is significantly higher for the CIFAR10 and MNIST datasets compared to random models, prompting us to test standard benchmarks.

Understanding the impact of a sign flip is key to determining the right balance between efficiency and correctness. For example, in a CIFAR10 model with 128 neurons and 8 hidden layers, keeping one hard-to-extract neuron sign flipped in layer 2 decreases the test accuracy to 0.9451 compared to the original model. With 5 sign flips, test accuracy falls between 0.55 and 0.68, and with 28 sign flips, it further declines to 0.268. These results illustrate that the effect of a sign flip is more significant than previously believed, suggesting that all prior methods of sign extraction may not have been adequate for consecutive layers' extractions, as none of them offered perfectly accurate sign extraction. The importance of accurate sign extraction is underscored by the failure of signature recovery in subsequent layers if even a single neuron's sign in a prior layer is incorrect.

### 3.3 Method for Correct and Efficient Extraction

Following the discovery that the confidence level and the number of correctly recovered neurons stabilises after about $s = 15$ iterations, we propose that sign extraction can be made more efficient by running it with less iterations. Further, we find that 'hard' to sign extract neurons' signs cannot be extracted with the Neuron Wiggle method. Given that the sign of some neurons cannot be determined, a new robust method for determining them must be sought. Our approach is to test possible values for these hard to extract signs in the next layer. Correct signs can be detected as they will allow signature extraction to complete in the next layer without an error. Conversely, incorrect sign values will result in an error and the signature process can be stopped and restarted with a different combination of neuron signs. This process can be streamlined further in following way:

(1) Hard to extract neurons are determined as those with low confidence (between 0.5 and 0.6 of true or false confidence since it is not known if true or false) after 15 iterations ($s = 15$).

(2) The next layer is extracted in parallel for each possible combinations of signs for these low-confidence neurons. If, for instance, there are five such neurons, $2^5$ signature extraction processes are initiated. At latest in the sign extraction an error is thrown for an incorrect signature extraction through a check described below. Each process with an error is terminated. Ultimately, only one error-free process should remain, ensuring accurate layer extraction without significantly extending execution time.

**Note:** When extracting one target neuron's sign, errors in the 'sample distance check' lead to having to rerun that iteration of the sign extraction with a new critical point. If an iteration has had to be rerun more than $10 - 20$ times with high probability it can be deemed that the signature extraction must have been erroneous. Since each iteration in the sign recovery costs 5 queries this will equate to about 100 queries until we have a strong indication that the signature extraction for at least one neuron was wrong. If we wanted to, this could be used to check the correctness of each neuron's

signature, by trying sign extraction for each in this way. If the sign extraction was wrong in the previous layer, all neuron signatures will be wrong. If the sign extraction in the previous layer was right but there was an error in the subsequent signature extraction, usually only one or two neuron signatures will be wrong. This check is also a good way to find these one or two erroneous signature extractions early on, so that the signature extraction can be rerun for these neurons before continuing with extraction in subsequent layers.

**The Sample Distance Check:** Specifically, the sample distance check fails continuously if the signature recovery is incorrect. The sign recovery compares the outputs $f(\mathbf{x}^*)$, $f(\mathbf{x}^* + \Delta)$, and $f(\mathbf{x}^* - \Delta)$ with the subtractions $sL = f(\mathbf{x}^* - \Delta) - f(\mathbf{x}^*)$ and $sR = f(\mathbf{x}^* + \Delta) - f(\mathbf{x}^*)$. If $||sL - sR|| \leq 10^{-13}$, then the process is aborted and restarts at a new critical point, as such a small difference suggests that the neuron's activation state may not have changed. This indicates a potential error in identifying a critical point. The threshold of $10^{-13}$ is used due to the precision limitations of `float64` being at about $10^{-15}$ and correct wiggles typically impacting the output in the order of $10^{-9}$. Consecutive failures suggest that there must be errors in the neuron's signature since the search for additional critical points in the sign extraction is guided by them.

### 3.4 Sign Extraction as Bottleneck?

According to Carlini et al. [1], the most time-intensive aspect of parameter extraction is sign extraction. Canales-Martínez et al. [2] enhance efficiency with the Neuron Wiggle method, reducing the operation complexity in a single layer to $O(sd_id^3)$, where $s$ is the number of critical points needed for sign recovery, and $d$ is the maximum of $d_0$ and $d_i$. Since Canales-Martínez et al. [2] only included the running time of their sign extraction in seconds without query numbers or a direct comparison to the signature extraction, from reading their paper, it appears as if the sign extraction is still the bottleneck in the parameter extraction. Running Carlini et al. [1]'s and Canales-Martínez et al. [2]'s codebase separately also underscores this discovery [ref. Figure 1(a) **CM** original]. However, comparing query numbers [ref. Figure 4 in Appendix C] and unifying the pipeline with Carlini's signature extraction to ensure comparability between signature and sign extraction time shows that contrary to prior results, sign extraction is not the bottleneck anymore [ref. Figure 1(a) **CM** unified]. In fact, with our adaptations it becomes the least time-significant part of parameter extraction. Please refer to Appendix C for more details on the implementation difference between **CM** original and **CM** unified.

### 3.5 Further Improvements

1. **Improvements in Signature Extraction:**
   (a) The process of finding partial signatures was improved to find specific missing partial signatures which diversify the view and help to construct the full signature.
   (b) The memory and running time was improved by discarding all partial signatures that do not contribute a new view to the full signature (memory deduplication).

2. **Improvements in Precision Improvement:** (The precision improvement function improves precision from `float32` to `float64`.)
   (a) The precision improvement function was adjusted to become usable for `MNIST` models.
   (b) The precision improvement function was identified as unnecessary to obtain signs. Sign extraction can be performed in `float16` or `float32` just as correctly.
   (c) The precision improvement function was identified as unnecessary for the next layer's signature and sign recovery if goal is the extraction of weights and biases to the precision of `float32` instead of `float64`. Moreover, if `float64` precision is needed this can run in parallel to extraction of the next layer.

Please refer to Appendix A for more details on these improvements.

## 4 Evaluation

### 4.1 Scalability and Accuracy in Neuron Sign Prediction

We assess the accuracy of neuron sign predictions on standard benchmarks by examining `MNIST` and `CIFAR` models with various configurations of hidden layers. We find that the number of low confident

and incorrectly identified neurons do not exceed 10. In this way, things should remain scalable, as parallelisation does not exceed more than $2^{10}$ signature extractions. More details are in Appendix B.

## 4.2 Performance

| Model Information | | Extraction Time [s] | | | | | Reduction | | |
|---|---|---|---|---|---|---|---|---|---|
| | | Signature | | Sign | | | Sign | Total | |
| Model | Params | C+CM | Ours | C | CM | Ours | CM→ Ours | C→ CM | CM→ Ours |
| 10-5x2-1 | 30 | 18.08 | 18.65 | 76.39 | 0.82 | 0.05 | ×16.40 | ×5.00 | ×1.01 |
| 20-10x2-1 | 110 | 13.38 | 13.17 | 86.38 | 1.59 | 0.12 | ×13.25 | ×6.66 | ×1.13 |
| 30-15x2-1 | 240 | 22.81 | 22.39 | 141.24 | 2.37 | 0.16 | ×14.81 | ×6.52 | ×1.12 |
| 40-20x2-1 | 420 | 27.59 | 27.96 | 193.52 | 3.46 | 0.28 | ×12.36 | ×7.12 | ×1.10 |
| 50-25x2-1 | 650 | 29.34 | 29.64 | $\approx 1.3 \cdot 10^5$ | 4.98 | 0.34 | ×14.65 | $\approx \times 3788.73$ | ×1.15 |
| 60-30x2-1 | 930 | 41.79 | 40.80 | $\approx 5.4 \cdot 10^6$ | 6.52 | 0.50 | ×13.04 | $\approx \times 1.1 \cdot 10^5$ | ×1.17 |
| 70-35x2-1 | 1260 | 107.70 | 46.15 | - | 10.58 | 0.77 | ×13.74 | - | ×2.52 |
| 80-40x2-1 | 1640 | 67.01 | 65.93 | - | 13.46 | 0.94 | ×14.32 | - | ×1.20 |
| 90-45x2-1 | 2070 | 96.28 | 94.37 | - | 18.61 | 1.41 | ×13.20 | - | ×1.20 |
| 100-50x2-1 | 2550 | 206.65 | 186.53 | - | 20.47 | 1.82 | ×11.25 | - | ×1.21 |

Table 1: Extraction Performance Carlini (**C**), Canales-Martinez (**CM**) versus **Ours** on layer 2 of random models. Since extraction times vary significantly between layers in different models, we perform comparison of layer by layer extraction time and not whole model extraction time. We compare layer 2 because layer 1 and 3 are more straightforward to extract. A 10-5x2-1 model, following Carlini et al. [1], represents input layer of size 10, two hidden layers of size 5 and output layer of size 1. The numbers highlighted in red capture the gist of the performance improvement and the numbers in blue are our best performances.

**Carlini vs. Canales-Martinez:** Table 1 illustrates the performance gains achieved by Canales-Martinez et al.'s (**CM**) extraction method and our extraction method compared to the previous version from Carlini et al. (**C**). The performance was tested on AMD Ryzen 7 4700U processor with 16GB RAM. **CM**'s sign extraction reduces the overall parameter extraction time in an exponential manner.

**Carlini + Canales-Martinez vs. Ours:** Moreover, our approach to parameter extraction has proven to be up to 16.40 times faster in the sign extraction and up to 2.52 times faster in the whole parameter extraction compared to Canales-Martínez et al. [2]. Only some of the signature extractions show significant improvements. This is because the enhancements in the critical point search process and memory deduplication (discarding irrelevant critical points) only improves performance in some cases. However, if as suggested we were to bypass precision improvement in the signature extraction, since this can run in parallel while already starting sign extraction and the subsequent layer's extraction, then higher performance improvement is possible. For the extractions in Table 1 the precision improvement makes up approximately $26\%, 82\%, 68\%, 59\%, 69\%, 61\%, 63\%, 53\%, 43\%$ and $45\%$ of signature extraction time respectively, resulting in an overall extraction speedup of $[1.36, 6.01, 3.44, 2.64, 3.60, 2.94, 6.63, 2.52, 2.08, 2.18]$ if precision improvement is disregarded.

Additionally, we have achieved performance gains in our sign extraction by setting the parameter $s$ to 15, utilizing knowledge about easy and hard to extract neurons. As mentioned in Section 3.5, the sign extraction can be performed equivalently in `float16`, `float32`, or `float64` without affecting correctness of sign extraction. While Table 1 shows results for `float32` sign extraction, empirically the performance for `float16` and `float64` is almost identical. The precise sign extraction time can change up to a few seconds if the signature extraction precision is lower, causing more errors to be thrown, in which case `float16` performs most stably.

**Efficiency Gains from Memory Deduplication:** Our analysis reveals significant improvements in signature extraction efficiency when implementing memory deduplication in the signature extraction, particularly in larger models. When examining the mean differences across four extraction seeds, we observed that for layer 2 of a random model with 128 neurons per hidden layer, the signature extraction process was 1.3 times more time-efficient, 1.2 times more query-efficient, and 1.2 times more memory-efficient with memory deduplication. For layer 2 of an `MNIST` model with 64 neurons it was on average 2 times as time efficient, 1.2 times as query efficient and 1.3 times as memory efficient. These findings underscore efficiency gains through memory deduplication, contributing significantly to reductions in memory usage and extraction time. A further graph on how the whole

parameter extraction scales for `MNIST` models with increasing layer sizes, and the calculation of the extraction time in the model used in the abstract can be found in Appendix C.

## 5    Discussion

| Model Information | | | Signature [s] | | Sign [s] | | Queries | |
|---|---|---|---|---|---|---|---|---|
| Model (Training Seed) | Layer | Params | Mean | Var | Mean | Var | Mean | Var |
| 784-8x2-1 (s1) | 2 | 72 | 10.39 | 0.25 | 0.25 | 0.002 | $5.13 \cdot 10^4$ | $4.9 \cdot 10^8$ |
| 784-16x2-1 (s1) | 2 | 272 | 7.22 | 8.85 | 0.60 | 0.005 | $6.92 \cdot 10^4$ | $9.3 \cdot 10^8$ |
| 784-32x2-1 (s1) | 2 | 1056 | 22.58 | 31.59 | 2.07 | 0.61 | $2.28 \cdot 10^5$ | $3.7 \cdot 10^9$ |
| 784-64x2-1 (s1) | 2 | 4096 | 135.32 | $2.9 \cdot 10^3$ | 7.17 | 6.32 | $9.03 \cdot 10^5$ | $1.9 \cdot 10^{10}$ |
| 784-128x2-1 (s1) | 2 | 16512 | 758.5 | $1.5 \cdot 10^5$ | 30.46 | 8.02 | $4.17 \cdot 10^6$ | $1.1 \cdot 10^6$ |
| 784-128x2-1 (s2) | 2 | 16512 | 1040.85 | 103.32 | 30.66 | 5.72 | $4.35 \cdot 10^6$ | $1.5 \cdot 10^6$ |
| | | | | | | | | |
| MNIST784-8x2-1 (s2) | 2 | 72 | 12.75 | 9.17 | 0.26 | 0 | 49,730 | $9.6 \cdot 10^5$ |
| MNIST784-16x2-1 (s2) | 2 | 272 | 19.15 | 37.03 | 0.67 | 0.01 | $1.92 \cdot 10^5$ | $4.6 \cdot 10^9$ |
| MNIST784-32x2-1 (s2) | 2 | 1056 | 98.10 | 1179.81 | 2.00 | 0.07 | $7.7 \cdot 10^5$ | $8.0 \cdot 10^{10}$ |
| MNIST784-64x2-1 (s2) | 2 | 4096 | 496.2 | $1.5 \cdot 10^5$ | 6.32 | 0.32 | $3.05 \cdot 10^6$ | $1.1 \cdot 10^{13}$ |
| MNIST784-64x2-1 (s1) | 2 | 4096 | 4649.95 | $1.6 \cdot 10^6$ | 6.85 | 1.79 | $4.9 \cdot 10^6$ | $2.8 \cdot 10^{13}$ |
| | | | | | | | | |
| MNIST784-16x8-1 (s2) | 1 | 12560 | $1 \cdot 10^4$ | - | 63.04 | - | $5.38 \cdot 10^6$ | - |
| MNIST784-16x8-1 (s2) | 2 | 272 | 470.19 | $3.4 \cdot 10^4$ | 0.67 | 0 | $5.27 \cdot 10^5$ | $1.2 \cdot 10^{10}$ |
| MNIST784-16x8-1 (s2) | 4 | 272 | $> 36hrs$ | | | | | |
| MNIST784-16x8-1 (s2) | 8 | 272 | $> 36hrs$ | | | | | |
| MNIST784-16x8-1 (s2) | 9 | 17 | 0.01 | 0 | 0 | 0 | 100 | 0 |
| | | | | | | | | |
| MNIST784-16x3-1 (s1) | 2 | 272 | 1854.42 | $2 \cdot 10^6$ | 0.96 | 0.15 | $9.7 \cdot 10^6$ | $5.2 \cdot 10^{13}$ |
| MNIST784-16x3-1 (s1) | 3 | 272 | $6.9 \cdot 10^4$ | - | 0.54 | - | $4.4 \cdot 10^7$ | - |

Table 2: Extraction Performance across models, training seeds and extractions seeds. The two different training seeds used are denoted as s1 and s2. The measurements were all taken over four extraction seeds. All signature extraction times are without the precision improvement function, since for `MNIST` models this takes up to 33 times longer than the actual signature extraction time and we have shown that this can be skipped or handled while already proceeding with further extraction processes. Extractions of deeper layers of MNIST784-16x8-1 did not lead to a full extraction after 36 hours with 6/16 and 0/16 extracted for layers 4 and 8. The most interesting contrasting results for discussion are highlighted pairwise in colours. In green one can see how layer 2 extraction for the same number of neurons can vary with model depth. In blue one can see the variance of extracting two models trained similarly but on different randomness. In red one can see how deeper layers become increasingly hard to extract.

### 5.1    Running Time of different models

In Table 2 an overview of different model extractions are presented. All extractions in this table were run on a High Performance Cluster with Intel's 10th generation Intel Core processors icelake. In the following, three cases are presented which show the limitations of the signature extraction:

**Case "Random" vs. `MNIST` Models:** "Random" models were used for testing by Carlini et al. [1]. These are not configured for a specific task but train 100 epochs on randomly generated data. The `MNIST` models' we additionally analysed have accuracies ranging from 0.67 to 0.94 for the two hidden layer models and is 0.91 for the 8 hidden layer model. In Table 2 one can see that extraction for these "random" models is much faster compared to extraction of `MNIST` models. A look into the kernel density estimate of the weights visualises that the concentration of weights near 0 is much higher for real world models trained on `MNIST` or `CIFAR10`. These models exhibit greater representation sparsity, meaning that a significant number of neurons are rarely activated. As a result, many neurons are grouped together in a compact region of the parameter space where activations are infrequent and close to zero, highlighting their minimal impact on the model's predictions. Consequently, locating specific coordinates for these seldom-activated neurons is challenging, as it is difficult to identify inputs that effectively trigger their activation.

**Case Deeper Models:** Deeper models are harder to extract. As can be seen in Table 2, comparing the extraction time of the second layer in a two hidden layer model and in an eight hidden layer model makes this apparent. Additionally, extraction of deeper layers is increasingly time consuming – in order to obtain the full signature of a neuron, a set of critical points that activates all previous layer's neurons must be found. Only then can the neuron be looked at from all perspectives, so that the full signature can be obtained. Carlini et al. [1] and Canales-Martínez et al. [2] mention increasing difficulty of extraction for deeper networks in the context of expansive networks. These networks feature inner layers with more neurons than the number of inputs they receive. If any hidden layer's expansion factor exceeds that of the smallest layer by too much, problems may arise.

**Case Random Seeds:** Although trained in the same way, the training seed seems to impact how well models can be extracted. Additionally, extraction seeds make a big difference in the efficiency of signature extraction. For example, up to $8$ hours runtime difference were noted for the extraction of `MNIST784-16x8-1` model's layer 2, when utilising different training and extraction seeds. Furthermore, a variance of just the extraction seed in a `MNIST784-64x2-1` model's layer 2 made a difference of $938$ seconds. Additional insight on variance can be seen in Table 2, where average model extraction time for models trained similarly but with different seeds can be as high as nine times and the query variance for different extraction seeds as high as $10^{13}$. In some cases specific seeds trigger incorrect extraction of some neurons. For example, for the extraction of `MNIST784-16x3-1` model's layer 2, extraction seeds 0, 10, 42, took over 1,000 seconds each and returned 4, 5, and 10 incorrectly extracted neurons, while for extraction seed 40 the extraction took only $54$ seconds and was fully correct. Hence, for replicability underlying randomness should be considered.

## 6    Conclusion

Unifying previous methods into one codebase has facilitated a more thorough benchmarking of cryptanalytical parameter extraction techniques. Increased performance and robustness was achieved through new insights into neuron characteristics such as the identification of harder neurons whose sign extraction cannot be robustly performed with the Neuron Wiggle technique. Instead, we find a way to check for incorrect extractions in the next layer's extraction. Consequently we propose that the next layer's extraction can be performed in parallel for all combinations of these neurons' signs and then incorrect extractions can be filtered out through this check later in the pipeline. Contrary to previous assumptions, we found that signature extraction presents a more significant bottleneck than sign extraction, prompting a reevaluation of focus areas in parameter extraction. Furthermore, our evaluation revealed discrepancies in extraction times across models, with models trained on random data proving easier to extract than those on structured datasets like `MNIST` due to representation sparsity. Notably, models with fewer than four hidden layers exhibited quicker extraction times, sometimes within one to two hours, whereas deeper models faced increased extraction difficulties. Highlighting the variability of extraction difficulty, we propose comprehensive benchmarking of model extraction methods considering factors such as the target model's training dataset, training method, layer size, depth, the specific layer targeted, along with the use of varying seeds to reflect the impact of randomness. To allow for comparison among implementations, the underlying ML framework, computer hardware, extraction time and query number are also important to note.

### Acknowledgments

We want to thank Ross J. Anderson for his contributions during the early stages of this project.

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

# A  Further Improvements explained

## A.1  Further Refinements in Signature Extraction

### A.1.1  Brief Explanation of Signature Extraction

Here we assume that we been able to find critical points, which are on the decision boundary of activating or deactivating a neuron in our target layer, we describe briefly the idea of how Carlini et al. [1]'s signature extraction works. To understand how the critical point search works in detail, please refer to Carlini et al. [1].

Let us first go into the case of signature recovery for a neuron $\eta_j$ in the first layer with corresponding critical point $\mathbf{x}^* \in \mathbb{R}^{d_0}$. We know that a critical point lies on the decision boundary for $\eta_j$ to turn on or off. So, if we move a tiny amount into some direction this will either turn the neuron active or inactive. Since we are actually in the space of hyperplanes, there are actually $d_0$ directions we can go to turn the neuron on or off. If we construct input queries such that we can find the gradients in each of these directions then this will reveal the value for the neuron. The gradient in each direction essentially quantifies the rate at which $\eta_j$'s output changes with respect to small shifts in the input. Therefore, understanding these gradients allows us to infer the weights that $\eta_j$ assigns to each dimension of the input space.

Mathematically, we construct the input queries $\alpha_{i,-} = \frac{\partial f}{\partial e_i}(\mathbf{x}^* - \varepsilon e_i)$ and $\alpha_{i,+} = \frac{\partial f}{\partial e_i}(\mathbf{x}^* + \varepsilon e_i)$ with $\varepsilon \in \mathbb{R}$ a small number and $\{\mathbf{e}_1, ..., \mathbf{e}_{d_0}\}$ a standard basis of $\mathbb{R}^{d_0}$. These are partial derivatives (gradients) in direction $e_i$. Either $\alpha_{i,-}$ or $\alpha_{i,+}$ will tip $\eta_j$ into an active state. If we subtract $\alpha_{i,-}$ from $\alpha_{i,+}$ then the gradient information for all neurons except for $\eta_j$ will cancel out, essentially revealing second order partial derivatives of $f$ at $x^*$ in direction $e_i$. This gives us a multiple of the value for weight $a_i$ of $\eta_j$. Repeating this procedure in $d_0$ directions will gives us a complete set of multiples of neuron coordinates $a_1, \ldots, a_{d_0}$. To obtain a comparable basis between the elements we normalise them by dividing $a_1, \ldots, a_{d_0}$ by some $a_k$, usually $a_1$. This now recovers the full signature $(\frac{a_1}{a_1}, \frac{a_2}{a_1}, \ldots, \frac{a_{d_0}}{a_1})$. Critical points associated to the same neuron produce identical signatures, whereas those related to neurons in different layers create different signatures. This distinction allows us to identify signatures corresponding to neurons in layer 1.

Signatures in consequent layers cannot be as easily found because the direct control of input in the blackbox model is lost, so changing the input in one dimension at a time will not work. Instead of $e_i$ a vector $\delta_k$ of length $d_0$ in some random direction is sampled from $\mathcal{N}(0, \varepsilon I_{d_0})$. Then we find the second order partial derivative $\{y_k\} = \{\frac{\partial^2 f(\mathbf{x}^*)}{\partial \delta_1 \partial \delta_k}\}$ for $k = \{1, ..., d_{i-1}\}$ in a similar fashion as before by adding and subtracting the $\delta_k$ vector to the critical point, computing its partial derivative and subtracting $\alpha_{i,-}$ from $\alpha_{i,+}$. Having found second order partial derivatives in $d_{i-1}$ directions, we construct a hidden matrix $\mathbf{H}$ which denotes the output up to the previous layer $i-1$ for each input $x^* + \delta_k$. This hidden matrix also reveals at which places the ReLU in previous layers has set some values to zero, turning neurons off. Now, solving for $a$ in the dot product $< \mathbf{H}, \mathbf{a} >= \mathbf{y}$ with the least squares approximation gives us the signature of a neuron. However, since ReLUs in previous layers might have already set some values of the input to zero, different critical points give varying partial signatures. These must be combined to reconstruct the actual full signature.

**Unification of Partial Signatures**

The output up to layer $j$, $f_{1..j}(x^*)$ for some critical point $x^*$, is likely to have some negative values, effectively turning the neuron off, so that only a partial signature can be recovered for neuron $\eta^*$ of layer $j + 1$. Hence, as already mentioned different partial signatures are combined to unify into one complete signature for a neuron. On average half of the neurons in the previous layer are negative and all points in the neighbourhood of $x^*$ will produce entries of zero in the partial signature at those indices. Now, given $x_1, x_2$ witnesses to critical points of neuron $\eta*$, imagine that $f_{1..j}(x_1)$ produces a partial signature with entries $t_1 \subset \{1, ..., d_j\}$ and $f_{1..j}(x_2)$ correspondingly $t_2 \subset \{1, ..., d_j\}$. Then, as long as $t_1 \cap t_2$ has at least one element a joint set $t_1 \cup t_2$ can be made. Assume $r_1$ to be the weight vector produced with $x_1$ with entries $t_1$ and $r_2$ the weight vector for $x_2$. Then both are solutions for the same matrix row $\mathbf{A}_i^j$, so should be equivalent at indices $r_1 \cap r_2$ up to some multiple: $r_1[t_1 \cap t_2] = c * r_2[t_1 \cap t_2]$. From this one can compute $c$ to recover $r_{1,2}[t_1 \cap t_2]$. This unification procedure at the same time is also a check whether $x_1$ and $x_2$ are truly witnesses for the same critical

point. Because if this fails then $r_1$ and $r_2$ are not scalable by one $c$ and hence not parallel to each other. This is also why it is helpful to have a set $t_1 \cap t_2 \geq 2$ for cross-checking.

**Computing the optimal scaling constant for unification**

If only a single reference vector $r_1$ is used for scaling this can lead to significant imprecisions. The normalization for each partial signature is calculated with $\hat{\mathbf{A}}^1_{i,j}/\hat{\mathbf{A}}^1_{i,k}$, where $k$ is usually set to 1. However, if for example $\mathbf{A}^1_{i,k} < 10^{-\alpha}$, where $\alpha \gg 0$ and other $\mathbf{A}_{i,k}$ values are considerably larger, then normalization will produce precision errors and the further adjustment to other partial signatures through rescaling with a constant $c$ according to a single reference vector will accumulate even more of them.

To mitigate these errors, Carlini et al. propose a more robust approach:

1. Compute partial signatures $r_1, \ldots, r_n$ for all witnesses found for a layer $l$.

2. Cluster these signatures into sets $\{S_j\}_{j=1}^{d_l}$. Signatures that are the same up to an error tolerance and scale should be grouped together. They are deemed to correspond to the same neuron.

3. For each set, determine the optimal unification constant that normalises the scale of all the different partial signatures within the set.

In this way the propagation of errors through scaling should be minimised.

**Graph Clustering of Neuron Vectors:** We utilise graph clustering to determine subsets $S$ of neuron vectors. Define each vector $r_m$ in cluster $n$ by its coordinate $r_m^a$. A graph $G = (V, E)$ is constructed where each vertex in $V$ represents a vector $r_i$, and an edge in $E$ connects two vertices if their vectors are sufficiently similar to suggest they belong to the same neuron. Specifically, vertices $r_i$ and $r_j$ are connected if the sum of indicators $\sum_k \mathbb{1}[|r_i^k - r_j^k| < \epsilon]$ exceeds $\log d_0$, suggesting close proximity in at least $\log d_0$ dimensions. Carlini et al. [1] suggest that an $\epsilon$ of $10^{-5}$ is effective.

To obtain the best scaling factor for a partial signature $a$ it is paired with another partial signature $b$ from the same neuron and aligned such that $r_a^i = r_b^i \cdot C_{ab}$ for as many indices $i$ as possible. The calculation of all possible scales between two partial signatures can be expressed in a matrix of ratios, where each entry is $M_{i,a,b} = r_a^i/r_b^i$. We then select $C_{ab} = \text{median}_i M_{i,a,b}$ as the scaling factor between signatures $a$ and $b$, with the standard deviation $e_{ab} = \text{stdev}_i M_{i,a,b}$ serving as the error estimate.

If unifying row $x$ with $a$ and then combining with $b$ preserves precision, then ideally $C_{ax} \cdot C_{xb} = C_{ab}$. If not, and the combined error $e_{ax} + e_{xb}$ is less than $e_{ab}$, the scaling factor $C_{ab}$ is updated to $C_{ax} \cdot C_{xb}$, optimizing the scale unification. This process is iterated until no further improvements are found, and the optimal dimension for unification is determined by $a = \arg\min_a \sum_b e_{ab}$, resulting in the finalised vector $C_a$.

### A.1.2 Refinements

There is a more targeted critical point search that is described in Carlini et al. [1]. This has been slightly adapted to ensure discovery of a diverse set of critical points for each neuron so that the partial signatures obtained are complementing each other to a full signature and not repeating the same partial signature. For this, the indices of the still missing parts of the full signature are tracked and they are used to guide subsequent critical point searches. Additional debugging now prevents infinite loops that had previously been found in some signature extraction processes. They had been initiated in the targeted critical point search.

Furthermore, we discovered that for a whole set of easy to find neurons the full signature is already obtained after few rounds of the graph clustering algorithm. Yet, the general, untargeted critical point search continues until at least three critical points are found for each neuron. This not only accumulates critical points necessary for neurons whose signature was not fully found yet, but also accumulates a lot of critical points for neurons where the full signature was already obtained. However, this is very inefficient for the graph clustering because the number of critical points and partial signatures that must be computed each iteration keeps increasing. Hence, we have implemented a memory deduplication technique that discards of critical points and their partial signatures if for a particular neuron the full signature has already been obtained. Furthermore, in the targeted critical

point search, critical points that merely repeat existing partial signatures and do not provide new insights are also discarded. In this way memory is saved to store unnecessary critical points and partial signatures and the extraction process is sped up by not having to graph cluster unnecessary partial signatures.

## A.2 Further Refinements in Precision Improvement

### A.2.1 Brief Explanation of Precision Improvement

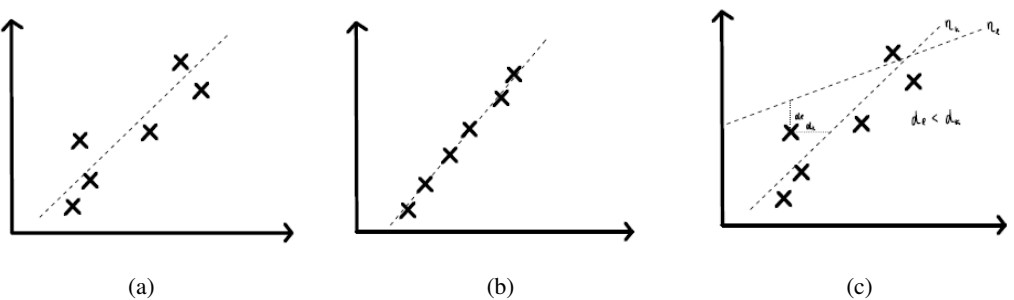

(a)                                        (b)                                        (c)

Figure 2: (a) Neuron and Critical Points before Precision Improvement (b) Neuron and Critical Points after Precision Improvement (c) An Example of When Precision Improvement Fails. Neuron $\eta_l$ is close to the critical point than $\eta_k$ and so this critical point is converted to a critical point for $\eta_l$ instead of for $\eta_k$.

In the signature extraction process small errors can be introduced in the calculation of the parameters which could accumulate to bigger errors in subsequent layers. Additionally, even if extraction is precise up to machine precision, the limited precision of `float64` can accumulate an error over time. This is why Carlini et al. [1] develop a method for improving the precision which brings the error from $2^{-15}$ down to $2^{-35}$ or lower. For each neuron $\eta_k$ in layer $j$, $j$ witnesses are computed by querying the up to layer $j$ extracted model $\hat{f}_{1\dots j}$, so that we obtain a set $\{x_i\}_{i=1}^{d_j}$ [ref. Fig. 2a]. For each $x_i$, if this is also a witness of the true model then its value $V(\eta; x_i) = 0$, if it is imprecise then the value will lie between 0 and some small error $\epsilon$. So, if we find that the value is still imprecise then we can take a random vector $\theta \in \mathbf{R}^{d_0}$ of small magnitude from the input space and find the true witness $x_i$ through a binary search in an interval $[x_i + \theta; x_i - \theta]$. Repeating this procedure for each $x_i$ will produce a true set of witnesses $\{x_i'\}_{i=1}^{d_j}$ [ref. Fig 2b].

Subsequently, the improved witness coordinates are put through the model up to layer $j - 1$ to obtain hidden vectors, $h_i' = \hat{f}_{1\dots j-1}(x_i')$. Since all previous layers are assumed to already be precise at this point, if $x_i'$ is truly a witness of $\eta_k$ then the output from layer $j$ for $\eta_k$ should be 0, i.e., $A_k^j * h_i' = 0$. Now, we try to fit a line through all critical points by solving with least squares error solution to the system of equations $H * x = [1\dots1]$, where $H$ is the hidden matrix made up of the hidden vectors [ref. Fig. 2b]. If the least squares solution error has gone down, the newly calculated neuron coordinate is swapped in for the previous coordinate. This process can be repeated until a certain precision is reached.

In practice, it could happen that in the conversion from $x_i$ to $x_i'$, accidentally the found $x_i'$ now is a witness to a different neuron. This is the case if a different neuron is closer to $x_i$ than $\eta_k$ [ref. Figure 2c]. Hence, the magnitude of $\theta$ becomes very important, as a too big value would contribute to this phenomenon but a too small $\theta$ would fail to find a new $x_i'$. Carlini et al. [1] set this to 0.1 to start with, as this was a value with which approximately half of the attempts at finding a new witness $x_i'$ failed. If too many solutions are found, $\theta$ is reduced to half and if no solutions are found $\theta$ is multiplied by 1.1. If the new solution is $A_k^j * h_i' < A_k^j * h_i$, then the new least squares approximation is set to be the new coordinate of $\eta_k$.

### A.2.2 Refinements

**Finding a true set of witnesses (critical points) and fitting a new line**

The precision improvement function struggled to identify critical points for `MNIST` models efficiently and often failed in the precision improvement function. Tackling the first part of the problem which is producing a true set of witnesses $\{x_i'\}_{i=1}^{d_j}$, several changes were made for enhanced correctness and efficiency.

The method for discovering critical points was refined. Critical points are calculated with a regression function that uses a loss function based on the magnitude of $A_k^j * h_i'$, where $h_i' = \hat{f}_{1\ldots j-1}(x_i')$. If $x_i'$ is truly a witness of $\eta_k$ then the output should equate to zero. Originally, the loss was calculated as the sum of all values in the zero vector, which occasionally led to suboptimal points where most values were near zero except one or two outliers. To address this, the maximum value of the vector was added to the loss, i.e., $\sum(A_k^j * h_i') + \max(A_k^j * h_i')$, ensuring all significant deviations are accounted for in the loss calculation. Additionally, previously identified critical points for $\eta_k$ are now included as starting points in the search for new witnesses. These points aid the optimization of the training process.

Refinements have also been made to the criteria for rejecting critical points computed. Initially, a critical point was rejected if the smallest activation in the target layer was smaller or equal to $90\%$ of the smallest activation in the target layer. In practice this, however, only rejected critical points if the minimum activation was equal to $0$. This was triggered endlessly for some models. To improve accuracy, the rejection now occurs if the maximum value $\max(A_k^j * h_i')$ exceeds $1e-5$, aligning with the approach used in the loss function.

Furthermore, in the second step, where a line is fitted through critical points to pinpoint a more precise coordinate for $A_k^j$, new coordinates with values near zero were causing inaccuracies and were thus rejected if any value fell below $10^{-5}$. This filtering process effectively eliminated problematic suggestions that previously impacted correct signature extraction.

**Do we need precision improvement?**

The improvements in the precision improvement enabled testing of this function for `MNIST` models, however, even with some changes to enhance efficiency, the precision improvement function turned out to be another bottleneck of the parameter extraction. After working correctly, the initial version took $34.7$ seconds on average per neuron for the precision improvement for a layer with $8$ neurons from a small `MNIST` model. Employing some of the tweaks decreased this to about $23.3$ seconds. However, this being $33$ times more than the signature recovery time and $17$ times more than the sign recovery time, still seemed like a major bottleneck. This raises the question: is such high precision necessary?

Currently, signature extraction achieves a precision between $10^{-8}$ and $10^{-10}$, which aligns with `float32` standards. The precision improvement function aimed to enhance precision to approximately $10^{-15}$, venturing into `float64` territory. Previously, we needed a `float64` accuracy to proceed with the sign extraction, as we assume the blackbox model to be in `float64`. However, experiments in which the extracted weights are changed to `float32` and even `float16` precision show that this does not significantly impact the accuracy of the sign recovery. In this setting the target blackbox model is still assumed to be in `float64` precision and the only things that are changed in the sign extraction are the precision of the extracted weights and biases and some parameters regarding epsilon values.

During sign extraction at these higher quantization levels, faults in the sample distance check start to become more frequent and sometimes the computation of the wiggle fails. Nevertheless, the sign extraction is still able to complete without becoming stuck in an infinite loop of error. Although recalculation of critical points is sometimes necessary due to sample distance check faults, both `float32` and `float64` showed similar query efficiency, with `float16` lagging due to more frequent errors. In terms of processing time, `float32` extraction proved faster than using `float64`, while `float16` was slower because of increased fault rates. Ultimately, we do not need to extract up to `float64` precision for the subsequent sign recovery and can therefore skip the precision enhancement function, making extraction a lot more efficient.

This raises the question of whether subsequent layers can effectively be extracted with prior layers at `float32` precision, and whether the following sign extraction processes remain functional. The answer is yes, the precision is often lower than if we had worked with previous layers extracted to `float64`, but it still extracts a correct signature. For example, for an `MNIST` model extracted with prior layers at `float32` precision the subsequent layer signature still extracts up to precision of

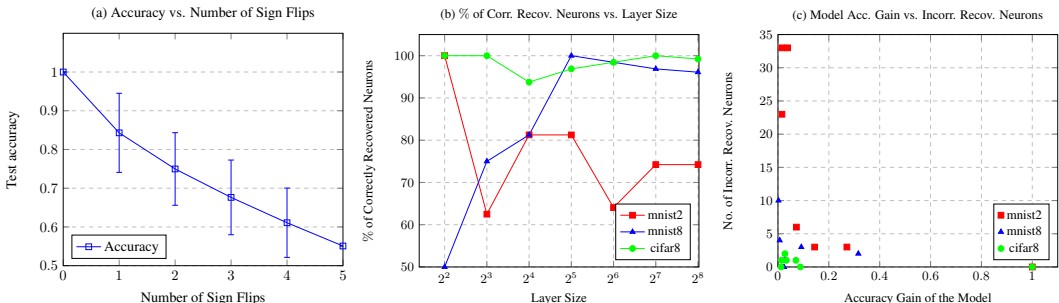

Figure 3: (a) The change of accuracy to original model's predictions with sign flips of hard to sign extract neurons in layer 3 of a `CIFAR` model with 128 neurons. The order of sign flipping was iterated over all combinations of ordering the 5 neurons to produce the error bounds. (b) Percentage of correctly recovered neurons in `MNIST` and `CIFAR` models with layer sizes ranging from 4 to 256. (c) Depicts how the number of incorrectly recovered neurons rises as the accuracy gain of a model due to larger layer size diminishes.

between $10^{-6}$ to $10^{-8}$, where if the prior layer was extracted up to `float64` precision the subsequent layer extraction precision was between $10^{-7}$ to $10^{-9}$ . Occasionally, the subsequent sign extraction in `float32` does not work and gets stuck on the sample distance check fault. However, employing `float16` settings in these instances circumvents these errors, allowing the process to complete correctly.

A remaining concern is that the precision will start decreasing more and more from layer to layer, so that deeper layers' signature extractions may start becoming infeasible. Yet, precision improvements for a previous layer can be computed in parallel with the start of extraction for subsequent layers. This approach ensures that once the precision of an earlier layer is enhanced to near `float64` levels, it can replace the earlier `float32` precision extraction, thus maintaining the integrity and feasibility of the overall extraction process.

## B   Scalability and Accuracy in Neuron Sign Prediction

Prior extraction performance evaluations have primarily utilised models trained on random data originally developed by Carlini et al. [1]. However, observations indicate that the percentage of correctly sign extracted neurons is significantly higher for models trained on standard benchmarks compared to random models. We assess the accuracy of neuron sign predictions on `MNIST` and `CIFAR10` models with various configurations of hidden layers. Specifically, we analyze `MNIST` models with either 2 or 8 hidden layers, and `CIFAR10` models with 8 hidden layers and layer sizes ranging from $2^2$ to $2^8$. To handle neurons with potentially incorrect signs, the next layer is extracted with all possible combinations of signs for these low-confidence neurons. The one error free extraction will confirm the correct signs for us (see Section 3.3). We assume that it is reasonable to handle up to 10 low-confidence neurons per layer and up to $2^{10}$ parallel signature extractions.

**The Number of Incorrect Neurons $< 10$?**

Our findings reveal that `MNIST` models with only 2 hidden layers exhibit a significantly higher number of incorrectly predicted neuron signs compared to those with 8 hidden layers, despite being trained under identical conditions [ref. Fig. 3(b)]. Given that most target models possess at least 8 hidden layers, this should be no problem in real life setting.

Furthermore, as shown in Figure 3(c), the number of incorrectly extracted neurons increases for models where adding more neurons to a layer did not significantly improve model accuracy. This plateau in performance indicates that for these models fewer neurons could have been sufficient for effective predictions. If we were to envision the neuron distribution of such a network, it is likely that many neurons would be positioned closely together, contributing to the decision-making process in a similar manner.

A structurally pruned network, which eliminates closely spaced redundant nodes, typically avoids these problems, placing neurons distinctly in the parameter space. This placement enhances sign

prediction accuracy. Consequently, the described sign extraction method with parallelisation in the subsequent layer's signature extraction should effectively scale with network size, since real-world models are most likely to exhibit these characteristics.

As shown in Figure 3, `CIFAR10` and `MNIST` models with 8 hidden layers have a maximum threshold of incorrectly predicted neurons of 10 in the `MNIST` model with hidden layer size 256. Yet, this hidden layer size is already unnecessary for the prediction of `MNIST` as can be seen in the graph since accuracy does not improve much. Taking this model out from our samples we are left with 13 models, where the maximum number of incorrectly recovered neurons is 4 if we run sign extraction $s = 15$ iterations for each model. These numbers underscore the methods real-world applicability. In fact, Canales-Martinez et al. [2] suggest that in high-dimensional spaces, the likelihood of two random vectors being perpendicular increases with the number of neurons in a layer, thereby enhancing the distinctiveness of the target neuron's wiggle. This property makes neurons predominantly straightforward to extract signs from, further affirming the method's robustness in real-world applications.

**Incorrect Neurons $\subseteq$ Low Confidence Neurons?**

Finally, we must determine if the incorrectly recovered neurons fall under the low-confidence category, defined by a confidence below $0.6$ with $s = 15$. In the sample of 13 models, most incorrectly extracted neurons were part of the low-confidence group. However, in 4 of the 13 models, there was an additional neuron incorrectly identified outside this group. Therefore, about $30\%$ of the time, another extraction round with $s = 15$ may be required. Typically, after a subsequent round, these previously misidentified neurons flip sign again and can be identified in this way. These neurons that experience a sign flip can then be included into the parallel signature extraction.

## C   Scalability of Whole Parameter Extraction

**Explanation of the runtime numbers in the Abstract:**   In the abstract we give the extraction time of a whole `MNIST` model of size 784-64x2-1 with 16,721 parameters. For both the original computation and our improved computation, the precision improvement function is excluded. In the original implementation the precision improvement function did not work for `MNIST` models and in our implementation the precision improvement function still took multiple times longer than other parts of the extraction. While we show that precision improvement is not necessarily needed, it would have been an unfair comparison to include this computational time only in the original computation. Furthermore, as shown in Section A.2.2, a precision of $10^{-8}$ is already obtained without employing the precision improvement function.

Additionally, we run both extraction computations under an optimal scenario, where prior layers' extractions are assumed to be correct. For example, if layer 3 is extracted, layer 1 and 2 are assumed to have already been correctly extracted. In practice, the method is not always that robust. For some extractions up to one or two neurons' signatures are sometimes extracted incorrectly and in the sign extraction both Canales-Martínez et al.'s and our version include some incorrectly sign identified low confident neurons (see Figure 1(b)). As suggested in Section 3.3, these errors can be found with the 'sample distance check'. However, this also means that in practice if errors are found, the signature extraction needs to be rerun. Similarly, for low confidence in sign extraction, $2^x$ signature extraction need to be run, where $x$ is the number of low confidence neurons.

We tested the extraction time over four extraction seeds. Extraction of layer 1 took approximately 138 minutes and 9,272,363 queries on average in the original Carlini+Canales-Martinez implementation and approximately 90 minutes and 5,157,296 queries on average in our implementation. Extraction of layer 2 took approximately 12 minutes and 4,630,078 queries on average in the original Carlini+Canales-Martinez implementation and approximately 8 minutes and 3,046,048 queries on average in our implementation. The final layer, layer 3, only takes 0.0008 seconds because a more direct method involving a system of linear equations can be used.

**Idealised Extraction:** Out codebase is built to extract weights and biases for one layer at a time following the example of Canales-Martínez et al.'s codebase. This was helpful for understanding the differences in extraction times for different layers. The extraction process becomes much more difficult for deeper layers but becomes very easy again for the last layer. Since extraction is performed layer by layer, previous layers are always assumed to be extracted correctly. Because of this idealised setting in our codebase, we have not actually implemented a parallelised signature extraction, since

this only becomes necessary if a prior layer's sign extraction was faulty. To get extraction times and query numbers for the whole extraction we have rerun the code for different layers separately, assuming that all prior layers' extraction were correct. To run the signature extraction on some next layer for all sign configurations of neurons with low confidence, the code for these can be run at the same time on different nodes.

Nevertheless, it is important to understand that if one executed all configurations in parallel, then while the extraction time stays on similar levels, the query numbers would be multiple times higher than they are for running signature extraction on just one configuration. In a parallelised version, queries in the general critical point search could be shared, but queries for the faster targeted critical point search could not be shared. Furthermore, while the easiest way to find an error was in the subsequent layer's sign extraction, further analysis of the signature extraction might reveal ways of finding faults earlier on that indicate the sign configuration in the prior layer was incorrect. Additionally, if the implementation considered the whole pipeline extraction, the critical points found that do not belong to the target layer could be saved to use in later layers. This could save us from having to execute some queries in the first part of the general critical point search for later layers.

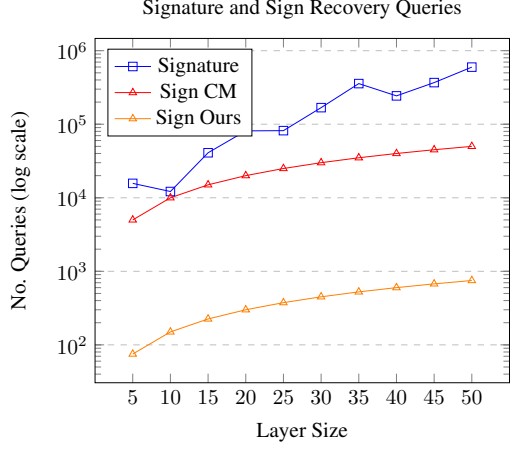

Figure 4: Compares the query numbers for Carlini's signature extraction versus Canales-Martinez (**CM**)'s sign extraction with $s = 200$ setting and Our sign extraction with $s = 15$ setting across ten models with increasing layer sizes from $10 - 5 - 5 - 1$ to $100 - 50 - 50 - 1$, detailing query numbers for a single layer's extraction.

| Model Information | | Extraction Time [s] | | | | | | |
|---|---|---|---|---|---|---|---|---|
| **Model** | **Params** | **Signature** | | **Sign (unified)** | | | **Sign (original)** | |
| | | **C+CM** | **Ours** | **C** | **CM** | **Ours** | **CM** | **Ours** |
| 10-5x2-1 | 30 | 18.08 | 18.65 | 76.39 | 0.82 | 0.05 | 156.54 | 6.00 |
| 20-10x2-1 | 110 | 13.38 | 13.17 | 86.38 | 1.59 | 0.12 | 184.63 | 13.16 |
| 30-15x2-1 | 240 | 22.81 | 22.39 | 141.24 | 2.37 | 0.16 | 254.06 | 18.55 |
| 40-20x2-1 | 420 | 27.59 | 27.96 | 193.52 | 3.46 | 0.28 | 317.57 | 23.93 |
| 50-25x2-1 | 650 | 29.34 | 29.64 | $\approx 1.3 \cdot 10^5$ | 4.98 | 0.34 | 439.40 | 30.50 |
| 60-30x2-1 | 930 | 41.79 | 40.80 | $\approx 5.4 \cdot 10^6$ | 6.52 | 0.50 | 478.91 | 36.16 |
| 70-35x2-1 | 1260 | 107.70 | 46.15 | - | 10.58 | 0.77 | 588.14 | 42.50 |
| 80-40x2-1 | 1640 | 67.01 | 65.93 | - | 13.46 | 0.94 | 667.79 | 48.68 |
| 90-45x2-1 | 2070 | 96.28 | 94.37 | - | 18.61 | 1.41 | 743.04 | 55.04 |
| 100-50x2-1 | 2550 | 206.65 | 186.53 | - | 20.47 | 1.82 | 844.72 | 61.21 |

Table 3: Extraction Performance Carlini (**C**), Canales-Martinez (**CM**) versus **Ours** on layer 2 of random models. The sign (unified) is the sign extraction time from the unified codebase, which has been used throughout the paper. The sign (original) time is the sign extraction time from the separate original codebase from **CM**. These are denoted here for completion.

**Explanation of Canales-Martínez et al.'s original versus unified implementation:** When unifying the codebases we noticed that Carlini et al.'s codebase had been written using jax as their machine learning library and their query calls were executed as a jax matrix multiplication of the weights and bias with the input. In contrast, Canales-Martínez et al.'s codebase was using TensorFlow and their query calls were executed through the predict() function in TensorFlow. To unify the codebase and ensure comparability between signature and sign extraction we chose to use jax as the basis machine learning library. This version, is the **CM** unified version that we use throughout the rest of the paper. The original and unified version do not exhibit any difference in query numbers or theory of the Neuron Wiggle sign extraction. Under **Sign (original)** in Table 3 one can see the sign extraction times with the separate original codebase from Canales-Martínez et al..

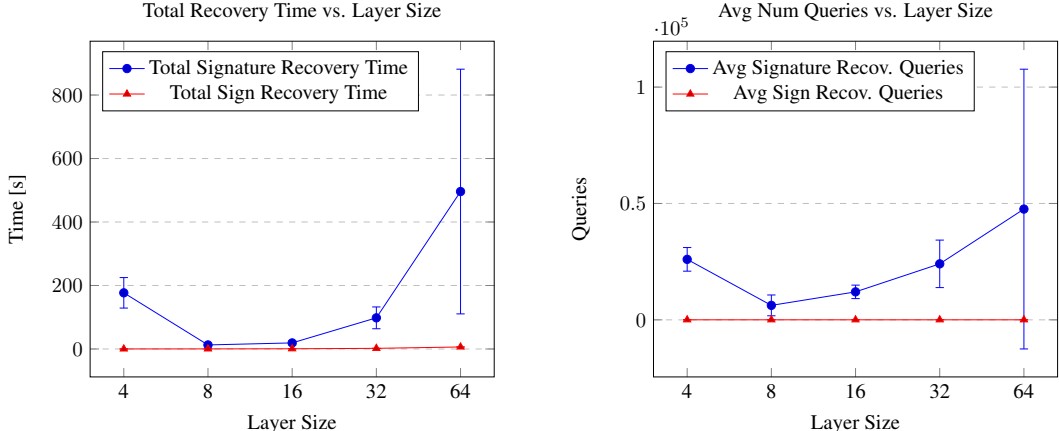

Figure 5: (a) Total signature recovery time and total sign recovery time of layer 2 of `MNIST` models with 2 hidden layers with layer sizes 4,8,16,32 and 64. The signature extraction was run with seeds 0,10,40 and 42. (b) Average number of queries for signature and sign recovery per neuron of layer 2 of `MNIST` models. These graphs do not include the precision improvement time or queries.

**Explanation of Figure 5:** Figure 5 shows how signature recovery and sign recovery for the second layer in `MNIST` models with layer structures ranging from $3072 - 4 - 4 - 1$ to $3072 - 64 - 64 - 1$ scale. The signature extraction was run with four different random seeds. In 5(a) we can see that the total sign recovery time is a lot smaller compared to the signature recovery time. This is due to two factors: First, the signature recovery takes longer in `MNIST` models compared to the random models we were looking at earlier. Second, the sign recovery takes a fraction of the time it took before.

