# OpenReview forum: "Beyond Slow Signs in High-fidelity Model Extraction"
_NeurIPS.cc/2024/Conference — NeurIPS 2024 poster_

### Official Review · Reviewer_aZHM · 2024-06-26

**Soundness:** 3
**Presentation:** 2
**Contribution:** 3
**Rating:** 6
**Confidence:** 2

**Summary:**

This paper proposes a unified approach to model parameter extraction by combining two prior works and integrating efficiency optimization. The authors find out that the neuron wiggle sign extraction method proposed in the previous work addresses the bottleneck identified by Calini's paper. Furthermore, the authors present methods to identify hard-to-extract neurons and parallelize the extraction process. Empirical evaluation of different benchmarks shows that the proposed unified model extraction method achieves speed up compared to prior works.

**Strengths:**

This paper has the following strengths:
+ The authors make an important observation that the neural wiggle method alleviates the sign extraction bottleneck in the prior work.
+ The authors show a unified model extraction method that combines sign extraction and model signature extraction built on top of two previous works.
+ The authors propose additional optimization methods to speed up the unified model extraction process.

**Weaknesses:**

This paper has the following weak points:
- There is no explicit description of the attack model/threat model used in this work;
- While the proposed unified model extraction method shows speedup compared to the prior works and the results are successful on standardly trained ML models, the benchmarks used in this paper are still small-scale. This makes the intellectual property value of the evaluated models questionable since they might be too simple and do not carry financial value.

**Questions:**

Please see the weak points above.

**Limitations:**

Please see the weak points above.

---

> ### Author Rebuttal · Authors · 2024-08-05
>
> > There is no explicit description of the attack model/threat model used in this work
>
> Currently, we have discussed the threat model in the background section, in part because it is the same as in the related work, namely, Carlini et al, Jagielski et al. Rolnick and Körding and Canales-Martinez et al.. Following feedback, we will add an explicit section on it.
>
> The threat model is an adversary aiming to replicate a model’s predictive capabilities from a black box setting where only the input and output are known to the adversary. The goal for the adversary is high fidelity, in other words the extracted model has to be as close to the original as possible. As described in Section 2.2 in Carlini et al. we assume following knowledge from the attacker: architecture knowledge, complete outputs, precise computations, scalar outputs and RELU activations. As described in Section 3 of Jagielski et al. the confidentiality of a model is compromised with a model extraction attack. Adversaries target the model for intellectual property and their goal is to extract a model with equivalent capabilities to the target model.
>
> This threat model is applicable to many machine learning models that are currently query accessible as ML-as-a-service, e.g. in the medical sector. Understanding limits of model extraction here is important, since developing these models requires vast datasets, extensive computational resources and expert knowledge, making them very valuable.
>
> > While the proposed unified model extraction method shows speedup compared to the prior works and the results are successful on standardly trained ML models, the benchmarks used in this paper are still small-scale. This makes the intellectual property value of the evaluated models questionable since they might be too simple and do not carry financial value.
>
> It is true that the attack is limited to models with limited depth and number of neurons per layer. However, we want to highlight that our paper uses the largest models compared to prior work. Our work further highlights that to scale to larger models, adversaries would need to improve on weight extraction methodology, since previously identified bottleneck is no longer relevant.
>
> Also, there exist small models that are valuable in this scale. These types of fully connected DNNs are used in practice e.g. in biology, healthcare, physics. For example, Smith et al. use a Multi Layer Perceptron with one hidden layer of size 500 for the classification of malignant childhood cerebellar tumors [1]. In a recent breakthrough advancing the field of nuclear fusion for sustainable energy, Degrave et al. [2] use such a fully connected DNN with 3 hidden layers of size 256 for the control policy of the tokamak plasmas. This is close to the size of models we tested and our attack improvements make stealing such a model feasible within a reasonable amount of time.
>
> [1] K. S. Smith et al., “Unified rhombic lip origins of group 3 and group 4 medulloblastoma,” Nature, vol. 609, no. 7929, pp. 1012–1020, Sep. 2022
>
> [2] J. Degrave, F. Felici, J. Buchli, M. Neunert, B. Tracey, F. Carpanese, T. Ewalds, R. Hafner, A. Abdolmaleki, D. de las Casas, C. Donner, L. Fritz, C. Galperti, A. Huber, J. Keeling, M. Tsimpoukelli, J. Kay, A. Merle, J.-M. Moret, S. Noury, F. Pesamosca, D. Pfau, O. Sauter, C. Sommariva, S. Coda, B. Duval, A. Fasoli, P. Kohli, K. Kavukcuoglu, D. Hassabis, and M. Riedmiller, "Magnetic control of tokamak plasmas through deep reinforcement learning," Nature, vol. 602, no. 7897, pp. 414–419, Feb. 2022

---

### Official Review · Reviewer_yp6m · 2024-07-09

**Soundness:** 3
**Presentation:** 3
**Contribution:** 3
**Rating:** 7
**Confidence:** 4

**Summary:**

This paper explores advanced techniques for high-fidelity model extraction that go beyond simply observing "slow signs" like model outputs or gradients. The authors evaluate and enhance existing parameter extraction methods, particularly those developed by Carlini et al. and further improved by Canales-Martínez et al., applying them to models trained on standard benchmarks. Key contributions include:
1. A unified codebase integrating previous methods
2. Optimizations to improve efficiency in extracting weight signs
3. Identification of weight extraction, rather than weight sign extraction, as the critical bottleneck
4. Significant improvements in extraction speed (e.g., extracting a 16,721 parameter MNIST model in 98 minutes vs. 150+ minutes previously)
5. Proposals for more robust benchmarking of model extraction attacks

**Strengths:**

- Comprehensive analysis and improvement of existing techniques
- Practical advancements in extraction efficiency
- Important insights into the relative difficulty of extracting different model components
- Contribution to standardizing evaluation methods in the field

**Weaknesses:**

- The scalability of the approach to larger, more complex models is not fully explored
- Potential ethical implications of improving model extraction techniques are not thoroughly discussed
- The paper could benefit from more discussion on potential defenses against these improved extraction methods

**Questions:**

1. Ethical Considerations: Given the potential misuse of advanced model extraction techniques, could you elaborate on the ethical implications of your work? What safeguards or guidelines do you propose to mitigate potential negative impacts?
Defense Mechanisms: While your paper focuses on improving extraction techniques, have you explored any potential defenses against these advanced model extraction attacks? If so, what were your findings, and if not, what are your thoughts on possible defensive strategies?

2. Defense Mechanisms: While your paper focuses on improving extraction techniques, have you explored any potential defenses against these advanced model extraction attacks? If so, what were your findings, and if not, what are your thoughts on possible defensive strategies?

**Limitations:**

The experiments are only conducted the ViT-B/32 CLIP architecture. It should be at least mentioned that in future it makes sense to justify on  more architectures.
The authors lowered this limitation by doing evaluation on several datasets.

---

> ### Author Rebuttal · Authors · 2024-08-05
>
> > Scalability of approach to larger, more complex models is not fully explored.
>
> With Table 2 we actually explore the largest models amongst all of the related work. Unfortunately the model extraction becomes very hard for deeper layers in for example an MNIST model with 8 hidden layers, so we had to stop attempting extraction for most of them. For models with more neurons per layer extraction also becomes increasingly harder and we were not able to finish extraction within 32 hours for layer size 128 MNIST models. In fact, our paper highlights that in order to make extraction work for even larger models we need further improvements to the weight extraction process.
>
> > Ethical Considerations: Given the potential misuse of advanced model extraction techniques, could you elaborate on the ethical implications of your work? What safeguards or guidelines do you propose to mitigate potential negative impacts?
>
> Ethically it is good for people to know about this vulnerability so they can (1) adjust their mental models of what adversaries are capable of, (2) adjust their machine learning models to not fit into the extraction criteria, and (3) to implement possible defenses. We will add a section to the paper that further goes into details on the above.
>
> > Defense Mechanisms: While your paper focuses on improving extraction techniques, have you explored any potential defenses against these advanced model extraction attacks? If so, what were your findings, and if not, what are your thoughts on possible defensive strategies?
>
> Current cryptanalytic model extraction only works for relatively small models with up to 3 hidden layers or so in practice, since extraction of deeper layers becomes harder since finding inputs that activate and deactivate a neuron in a later layer is harder due to deactivations of neurons in prior layers. So, larger models are currently not at risk. Furthermore, this extraction does not work with integer quantized weights since triggering an individual neuron is often not possible in this setting. Hence, smaller models quantized to integer weights can also not be extracted. This in no way provides any guarantees against other types of extraction as is mentioned in the background section e.g. with model distillation.
>
> **Query Accessibility:** Furthermore, for remaining models, query accessibility is needed. Millions of queries must be performed in order to extract a model. Companies could ratelimit the queries, so that extraction becomes harder. Even if adversaries were to be querying from a huge number of accounts, extraction would become a lot slower and easier to detect due to a huge increase in queries compared to usual operations, in line with the stateful detection work for example from Chen et al. [1].
>
> **Attack Prevention with Noise:** Adding noise to the output to limit the accuracy of the output should significantly limit the accuracy of cryptanalytic model extraction. However, perhaps instead one could then query ten times for each input and then take the average as the value to try signature extraction with. Ultimately, the minimum magnitude of the noise that would prevent the attack would depend on what precision extraction still works. Currently, signature extraction works for weights of float32 precision but could perhaps be tuned to work for float16 precision.
>
> **Attack Prevention with Parameter Variance:** We have found that models with minimized parameter variance across a layer are more resilient against this type of extraction attack because it is harder to distinguish between neurons and hence harder to find each neuron’s distinctive features. Hence, one could scale down parameters of a model or confine parameters of each layer in a different area of the parameter space so that extraction becomes increasingly difficult. Hence, our paper currently discusses high variance in performance and highlights the importance of the hyperparameters involved for a fair evaluation.
>
> We will add a subsection discussing these mitigations.
>
> [1] S. Chen, N. Carlini, and D. Wagner, “Stateful Detection of Black-Box Adversarial Attacks,” Proceedings of the 1st ACM Workshop on Security and Privacy on Artificial Intelligence, pp. 30–39, Oct. 2020
>
> > The experiments are only conducted the ViT-B/32 CLIP architecture. It should be at least mentioned that in future it makes sense to justify on more architectures. The authors lowered this limitation by doing evaluation on several datasets.
>
> We are not sure what this comment refers to. Our paper only looks at MLPs and ViTs are not mentioned in the manuscript at all. The datasets considered are standard for this branch of literature.

---

> > ### Comment · Reviewer_yp6m · 2024-08-13
> >
> > Thank you for answering my questions. I will leave my review unchanged.
> >
> > Note: I understand that sharing code in this field is uncommon. However, I believe that having a unified codebase could advance this research toward more realistic models and real-world scenarios.

---

> > > ### Author Response · Authors · 2024-08-13
> > >
> > > Thank you very much!
> > >
> > > We also do believe sharing code is important, it all lives here: https://anonymous.4open.science/r/anonymized-cryptanalytical-extraction-main-9335/README.md

---

> > > > ### Comment · Reviewer_yp6m · 2024-08-14
> > > >
> > > > It is convenient in the ML community to put the code after the abstract or introduction.

---

### Official Review · Reviewer_PoZg · 2024-07-24

**Soundness:** 2
**Presentation:** 2
**Contribution:** 2
**Rating:** 5
**Confidence:** 2

**Summary:**

This paper proposes to perform model extraction attacks against deep neural networks. First, the authors combine two previous proposed methods into a uniformed code-base. Then, they optimize the sign extraction strategy to achieve a speed up in model extraction. Their proposed method can be used for larger models (e.g., a model with 16721 parameters and two hidden layers).

**Strengths:**

- An important problem.

- The results show some efficiency of the proposed method.

**Weaknesses:**

- This paper is hard to understand. The main novelty and the insight behind it are unclear to me.

    - If Canales-Martinez’s work already eliminates the sign extraction bottleneck (Line 47), why the authors still propose to optimize this process?

    - What is the design motivation? Why the authors could do better?

    - Carlini’s work is not the only crypt-analytical work, why not compare to other works (e.g., reference 7)?

- Too much background in the Methodology part. For example, the whole Sec. 3.1 is about some background.

- Section 3.4 is hard to follow.

- The cost of parallel computing is exponential.

**Questions:**

See above.

**Limitations:**

See above.

---

> ### Author Rebuttal · Authors · 2024-08-05
>
> > Too much background in methodology, i.e. whole of Sec. 3.1
>
> Beginning from line 201 “Confidence in Practice” is our contribution. We will mark it more explicitly to differentiate clearer between prior contribution and our contribution.
>
> > Section 3.4 hard to follow
>
> We are very sorry to hear that the reviewer found Section 3.4 hard to follow, could you possibly clarify which bits were confusing? Reviewer yp6m scored the paper good for presentation and r46i even complimented the paper for distilling complex attacks in a readable way, and further called the paper very well written.
>
> > Cost of parallel computing is exponential
>
> Although the worst case complexity may be exponential, we find that in practice such worst case branching never happens, since detection for erroneous branching gets detected within the next layer in a handful of queries as we note in Section 3.2.
>
> > If Canales-Martinez’s work already eliminates the sign extraction bottleneck (Line 47), why the authors still propose to optimize this process?
>
> We did not know that Canales-Martinez et al.’s work already eliminated the sign extraction bottleneck because they did not test the whole extraction and the numbers from Carlini’s codebase and Canales-Martinez’ codebase were not comparable at first. Hence, the paper is written in a way to highlight that the previously identified bottleneck is no longer relevant and suggests that future improvements need to be made to the weight extraction process. What is more, it argues for a more thorough and detailed attack variance performance.
>
> > What is the design motivation? Why the authors could do better?
>
> We identified that some neurons are harder to extract than others and performing the neuron wiggle method on them for more iterations does not help extract more correct signs. So, overall we minimized the neuron wiggle extraction time for easy to extract neurons and performed sign extraction for hard to extract neurons later in the pipeline. In particular, we can identify hard to extract neurons and can identify through the signature and sign extraction in the next layer if the sign extraction for a neuron was wrong. In this way we also add robustness in extraction which was missing in Canales-Martinez et al.’s work. Additionally we identified that the precision improvement process was taking a long time especially for MNIST models.
>
> > Carlini’s work is not the only crypt-analytical work, why not compare to other works (e.g., reference 7)?
>
> We do not for example compare to reference 7, Rolnick and Körding (2020), because Carlini et al. already compare their work against them in their Table 1. Carlini et al. mentions that their work builds upon previous works including Milli et al. (2019), Jagielski et al. (2020), and Rolnick and Körding (2020). After developing their attack Carlini et al. compare against Rolnick and Körding (2020) to discover that Rolnick and Körding’s attack is significantly more query intensive, especially for extraction of more than the first two layers. Ultimately Carlini et al. claim that their method is 100+ times more accurate, query intense and can handle larger models than Rolnick and Körding (2020).

---

> > ### Author Response · Authors · 2024-08-12
> >
> > Dear Reviewer PoZg,
> >
> > Thank you so much for your insightful feedback on our paper! We hope you found our responses useful. As the discussion period is coming to a close, please feel free to ask any remaining questions you may have. We're happy to provide further clarification!

---

> > ### Comment · Reviewer_PoZg · 2024-08-14
> > **Thanks for the response**
> >
> > 1. Section 3.1's majority of contents are still backgrounds (about two pages, which occupies about half of the methodology section).
> > 2. Section 3.4 is unclear about the acceleration improvements that have been made. This part seems to focus on engineering efforts rather than systematic methodologies.
> > 3. According to the introduction and the methodology, the paper focuses most of its effort on sign extraction. Based on the rebuttal, I understand this paper tries to convince others that sign extraction is not a bottleneck. Is that correct? If so, why do the authors still spend effort on improving sign extraction (see abstract lines 12-14)?
> > 4. The authors stated "Hence, the paper is written in a way to highlight that the previously identified bottleneck is no longer relevant and suggests that future improvements need to be made to the weight extraction process." **However, according to the rebuttal, the paper's main contribution is to optimize the sign extraction attack instead of the weight extraction process.** "We identified that some neurons are harder to extract than others and performing the neuron wiggle method on them for more iterations does not help extract more correct signs. So, overall we minimized the neuron wiggle extraction time for easy to extract neurons and performed sign extraction for hard to extract neurons later in the pipeline." This is contradictory.
> >
> > Thanks again for the authors's response. However, my questions are not well-addressed. I decide to keep my score.

---

> > > ### Author Response · Authors · 2024-08-14
> > > **Thank you for your response**
> > >
> > > 1. The first part of Section  3.1 is indeed about one and a half pages of explaining previous works’ methodologies but they were necessary to make our contribution understandable. From reviewer r46i we received positive comments regarding this: “The paper is very well-written and does a great job of summarizing the fairly complex attacks that it builds on. Honestly this paper is one of the best I've ever seen for how well it values the reader's time and presents the relevant background and contributions.”
> > >
> > > 2. Section 3.4 is a mix of methodological improvements and engineering efforts. We could not go into detail about it in the paper due to limited space but have a full description of what these entail in Appendix A. Especially our insights on how precision improvement is not necessarily needed are not trivial, since this was not possible yet in Carlini et al.’s paper and Canales-Martinez et al. never connected signature and sign extraction.
> > >
> > > 3. It was not clear at first that sign extraction was not the bottleneck anymore until we unified all methodologies and metrics. In Figure 1 (a), we show with Sign CM original and unified compared to Signature Carlini that sign extraction was more inefficient before but after unifying codebases it is taking a little less time than the signature extraction. However, this does not mean sign extraction time became trivial just using Canales-Martinez et al.’s methods. Only with our improvements in sign extraction, the sign extraction actually becomes trivial compared to the signature extraction. Further, in the full pipeline our improvements in the sign extraction significantly improve the overall extraction time as shown in abstract lines 15-17.
> > >
> > > 4. We are not too sure what the reviewer means by contradictory, we followed prior work and improved over the sign extraction performance. By the time our improvements became significant we decided to evaluate the system, including evaluation of the system as a whole, not separately as Canales Martinez. This is where we realized that sign extraction time can become trivial compared to signature extraction, and future efforts should be directed towards signature extraction improvement. Note that everyone working in this field from reading all of the work assumed that sign extraction is the bottleneck and hence iterated over it.

---

> > > > ### Comment · Reviewer_PoZg · 2024-08-14
> > > > **Thanks again for the response.**
> > > >
> > > > 1. In my opinion, the background part should be included in the paper but go to the related work or background part. More space should be spent on technical contributions (e.g., Section 3.4).
> > > > 2. I acknowledge the authors' effort in justifying that sign extraction is no longer a bottleneck; this is indeed important. However, this is not so clear in the paper. I think the author would like to make them clearer in the paper.

---

> > > > > ### Author Response · Authors · 2024-08-14
> > > > > **Thank you for your prompt response.**
> > > > >
> > > > > Thank you very much for a prompt response!!
> > > > >
> > > > > 1. We will revise the structure to focus more on our technical contributions. We will put the background methodology in a specific background methodology subsection and will move some of it into the appendix. We will instead move some of our technical descriptions for Section 3.4 into the main paper methodologies.
> > > > >
> > > > > 2. We will work on clarifying this further.

---

### Official Review · Reviewer_r46i · 2024-07-24

**Soundness:** 3
**Presentation:** 3
**Contribution:** 3
**Rating:** 7
**Confidence:** 3

**Summary:**

This paper continues a line of work on cryptanalytically extracting network parameters from (input, logit) pairs. It includes a concise explanation of relevant prior work in the area, a codebase that unifies two key prior works in the area to enable standardized comparisons, and several improvements to these prior methods that enable faster extraction of weight signs and signatures.

**Strengths:**

- This is an interesting research area that deserves more attention from the broader community
- This paper not only makes progress in this area, but it also enables future work with its codebase
- The paper is very well-written and does a great job of summarizing the fairly complex attacks that it builds on. Honestly this paper is one of the best I've ever seen for how well it values the reader's time and presents the relevant background and contributions.
- The performance improvements are substantial
- The discussion includes some interesting analysis

**Weaknesses:**

Moderate issues:
- The main reason I'm not giving this paper a higher score is that it seems to make solid incremental improvements to two prior techniques. It doesn't introduce fundamentally new methods as far as I can tell. However, I think it still adds considerable value to the area, passes the quality bar, and could foster useful discussion at NeurIPS.

Minor issues / suggestions:
- My first question coming into this paper (not having read the prior work) was, "How on earth do people extract the layers one by one when all they have are the inputs and logits?" It took some time to understand this, and I think this is possibly something that could be improved with a well-designed figure.
- Line 85: "In a study where the adversary is assumed to have complete access to both the training data and hyperparameters, 93.4% was the maximum fidelity reached by the replicated model." What dataset is this on? Is 93.4% lower than expected? By how much?

Typos:
- Missing a space in the Figure 3 caption: "(a)The"

**Questions:**

Line 48 of the paper states, "Further improving on sign extraction we speed the process up by up to 14.8 times, so that sign extraction only takes up as little as 0.002% of the whole extraction time for larger models." A 14.8x speedup is nice, but doesn't this mean that sign extraction only took ~0.02% of the whole extraction time before? If so, then it was already negligible. Maybe this is part of your point about Canales-Martinez incorrectly estimated the difficulty of full extraction relative to sign extraction alone.

Which of the analyses in Section 3.1 are novel, and which have been touched on before in prior work?

Is there any chance that attacks like this could work for networks using smoother nonlinearities like GELU?

**Limitations:**

Adequately addressed

---

> ### Author Rebuttal · Authors · 2024-08-05
>
> > Line 85: "In a study where the adversary is assumed to have complete access to both the training data and hyperparameters, 93.4% was the maximum fidelity reached by the replicated model." What dataset is this on? Is 93.4% lower than expected? By how much?
>
> Jagielski et al. use the Fashion-MNIST dataset. They use this as an oracle labeled dataset and produce model f1 and change sources of determinism and run the learning-based attack to produce f2. They obtain 93.7% fidelity when training and initialization randomness are fixed and only GPU non-determinism remains. When no randomness is fixed, they obtain 93.4% fidelity. 93.4% fidelity is lower than expected given the oracle access to the labeled dataset. We will rephrase the sentence to make it more readable. It is also worth noting that there is now more recent work by Martinelli et al. [1] that suggests that high fidelity can actually be reached with learning based methods but it assumes a slightly different setting, only works for smaller networks, and appears to be more expensive.
> [1] F. Martinelli, B. Imşek, W. Gerstner, and J. Brea, “Expand-and-Cluster: Parameter Recovery of Neural Networks,” Arxiv, Apr. 2023. Accessed: Jul. 31, 2024. [Online].
>
> > Line 48. Line 48 of the paper states, "Further improving on sign extraction we speed the process up by up to 14.8 times, so that sign extraction only takes up as little as 0.002% of the whole extraction time for larger models." A 14.8x speedup is nice, but doesn't this mean that sign extraction only took ~0.02% of the whole extraction time before? If so, then it was already negligible. Maybe this is part of your point about Canales-Martinez incorrectly estimated the difficulty of full extraction relative to sign extraction alone.
>
> The 14.8 times speed up value comes from Table 1 extraction of a random model 30-15x2-1. For this model the total speedup was 1.12 compared to Canales-Martines et al.The 0.002% of whole extraction time comes from an MNIST model in Table 2 MNIST784-64x2-1 (s1). This is more to show that since signature extraction time has no bound, if signature extraction due to randomness takes a lot longer than sign extraction, signature extraction time can become very insignificantly small. However, for other models where signature extraction time is relatively small depending on reasons mentioned in the Discussion, the sign extraction time will weigh in more.
> Indeed, it is true that Canales-Martinez incorrectly estimated the difficulty of full extraction relative to only the sign extraction, so to start off with we did not know how sign extraction time compares to signature extraction time hence started looking into further optimizing the process.
>
> > Section 3.1: Which of the analyses in Section 3.1 are novel, and which have been touched on before in prior work?
>
> The part beginning from line 201 “Confidence in Practice” is novel. The parts before have been touched upon in prior work by Carlini et al. and Canales-Martinez et al.
>
> > Is there any chance that attacks like this could work for networks using smoother nonlinearities like GELU?
>
> According to Rolnick and Körding “Other piecewise linear activation functions admit similar algorithms.” In their Section 5 they further discuss other architectures and how well this type of cryptanalytical attack generalizes for them. We did not investigate this further since it is not the commonly assumed setting for this literature, but it is an interesting question for future work.
>
> > Suggestion: My first question coming into this paper (not having read the prior work) was, "How on earth do people extract the layers one by one when all they have are the inputs and logits?" It took some time to understand this, and I think this is possibly something that could be improved with a well-designed figure.
>
> We will make sure to work on improving readability and will look into adding a figure for this as per your suggestion.

---

### Decision · Program_Chairs · 2024-09-25

**Decision:**

Accept (poster)

**Comment:**

I agree with the unanimous decision of the reviewers to accept the paper. However, there has been some discussion regarding its presentation. I encourage the authors to consider the reviewers' feedback to enhance the paper's readability.